# Interactome overlap between risk genes of epilepsy and targets of anti-epileptic drugs

Yu-Qin Lv[1]☯, Xing Wang[1]☯, Yu-Zhuang Jiao[2], Yan-Hua Wang[1], Na Wang[3], Lei Gao◐[4]*, Jing-Jun Zhang[5]*

1 School of Clinical Medicine, Shandong First Medical University & Shandong Academy of Medical Sciences, Jinan, Shandong, China, 2 Shandong Provincial Qianfoshan Hospital, First Affiliated Hospital of Shandong First Medical University, Jinan, Shandong, China, 3 Department of Internal Medicine, Taishan Vocational College of Nursing, Tai'an, Shandong, China, 4 Department of Bioinformatics, School of Life Sciences, Shandong First Medical University & Shandong Academy of Medical Sciences, Tai'an, Shandong, China, 5 Department of Neurology, The second Affiliated Hospital of Shandong First Medical University, Tai'an, Shandong, China

☯ These authors contributed equally to this work.
* jjzhang63@126.com (JJZ); gaolei_tsmu@163.com (LG)

## Abstract

Anti-epileptic drugs have been used for treating epilepsy for decades, meanwhile, more than one hundred genes have been identified to be associated with risk of epilepsy; however, the interaction mechanism between anti-epileptic drugs and risk genes of epilepsy was still not clearly understood. In this study, we systematically explored the interaction of epilepsy risk genes and anti-epileptic drug targets through a network-based approach. Our results revealed that anti-epileptic drug targets were significantly over-represented in risk genes of epilepsy with 17 overlapping genes and P-value = $2.2 \times 10^{-16}$. We identified a significantly localized PPI network with 55 epileptic risk genes and 94 anti-epileptic drug target genes, and network overlap analysis showed significant interactome overlap between risk genes and drug targets with P-value = 0.04. Besides, genes from PPI network were significantly enriched in the co-expression network of epilepsy with 22 enriched genes and P-value = $1.3 \times 10^{-15}$; meanwhile, cell type enrichment analysis indicated genes in this network were significantly enriched in 4 brain cell types (Interneuron, Medium Spiny Neuron, CA1 pyramidal Neuron, and Somatosensory pyramidal Neuron). These results provide evidence for significant interactions between epilepsy risk genes and anti-epileptic drug targets from the perspective of network biology.

## Introduction

Epilepsy is a collective term for a group of syndromes caused by abnormal discharges of the nervous system, and can cause varying degrees of damage to behavior, cognition, and memory. Previous studies have shown that epilepsy genetic factors contributed a lot to the pathogenesis of epilepsy. Approximately 20–30% of epilepsy cases are caused by acquired conditions such as stroke, tumor or head injury, but there are still 70–80% of cases are considered to be associated with one or more genetic factors [1]. During these decades, whole exome sequencing [2],

**Data Availability Statement:** DrugBank:https://go. drugbank.com/; Interactome data and Network Calculator: https://github.com/Haoxiang-Qi/ Network-Calculator.git; EWCE and KI dataset:

https://github.com/NathanSkene/EWCE; http://www.hjerling-leffler-lab.org/data/scz_singlecell/.

**Funding:** This research was supported by Medical Health Science and Technology Project of Shandong Provincial Health Commission (2019WS391), Academic Promotion Program of Shandong First Medical University (2019QL013), National Natural Science Foundation of China (32000477), Shandong provincial Natural Science Foundation (ZR2020MC061), Science and Technology Program of Colleges and Universities in Shandong Province(J15LL07) and the planned project of Tai'an Science and Technology Bureau (2017NS0248).

**Competing interests:** The authors have declared that no competing interests exist.

genome-wide association studies [3], as well as researches using next generation sequencing technology [4] have identified variations of genes such as sodium channel, potassium channel and $GABA_A$ receptor that were clearly associated with multiple epileptic phenotypes [5].

Among treatments for epilepsy, anti-epileptic drug therapy are the most commonly used, in which most drugs exert their effects by regulating excitatory-inhibitory balance of the brain. Currently, more than one hundred of drug targets, which are involved of voltage-gated ion channels [6], γ-aminobutyric acid energy transfer [7], as well as glutamate energy transfer have been identified. However, the response of patients with epilepsy to anti-epileptic drugs varied greatly, which were largely due to genetic variations that affected both pharmacokinetics and pharmacodynamics of anti-epileptic drugs and risk of epilepsy [8]. Although a variety of drug targets and risk genes of epilepsy have been identified, the interaction between anti-epileptic drugs and risk genes of epilepsy were still not fully understood. With the development of system biology and the accumulation of interactome data, network biology has become an effective approach to explore the underlying biological mechanism of diseases by protein-protein interaction (PPI) networks [9], therefore, in this study, we implemented a network-based approach to explore the interactions between risk genes of epilepsy and anti-epileptic drug target genes. Meanwhile, emerging advances of single-cell RNA sequencing (scRNA-seq) in the central nervous system (CNS) have provide exciting molecular insights into understanding the complexity of the brain, as well as disease-relevant mechanisms by identifying novel cellular subtypes [10]. By applying knowledge of the cellular taxonomy of the brain from single-cell RNA sequencing, previous researchers have performed genetic identification of brain cell-types in schizophrenia [11]. In our study, we explored the molecular mechanism and cellular localization of the interaction network between anti-epileptic drug targets and epileptic risk genes by combining single cell sequencing data and network biological analysis. Our analysis may provide insights for understanding of the genetic basis of epilepsy and the development of anti-epileptic drugs.

## Materials and methods

The flowchart of our study was shown in Fig 1.

### 1. Identification of risk genes of epilepsy

We obtained risk genes of epilepsy identified by both common variants and rare variants, of which common variants are extracted from the largest trans-ethnic meta-analyses of genome-wide association studies currently [12], which included 15,212 individuals with epilepsy and 29,677 controls and identified 16 genome-wide significant loci with P-value$<5.0\times10^{-8}$. Rare variants were identified by exome-sequencing under the largest sample size of 1165 cases and 3877 controls, and their mapped genes were considered as monogenic epilepsy genes [12, 13]. Besides, we also included genes supported by literature retrieval and comprehensive databases providing genetic evidence of risk genes in disease (DisGeNET database [14] and MalaCards database [15]).

### 2. Identification of anti-epileptic drugs targets

DrugBank (https://go.drugbank.com/) is a web-based database containing comprehensive molecular information about drugs, their mechanisms of action, interactions, and their targets [16]. To obtain target genes of anti-epileptic drugs, we searched DrugBank database (Version 5.0) with ATC classification (N03 for anti-epileptic drugs), and a total of 47 anti-epileptic drugs with supported publications were collected. Fisher's Exact Test were performed with R

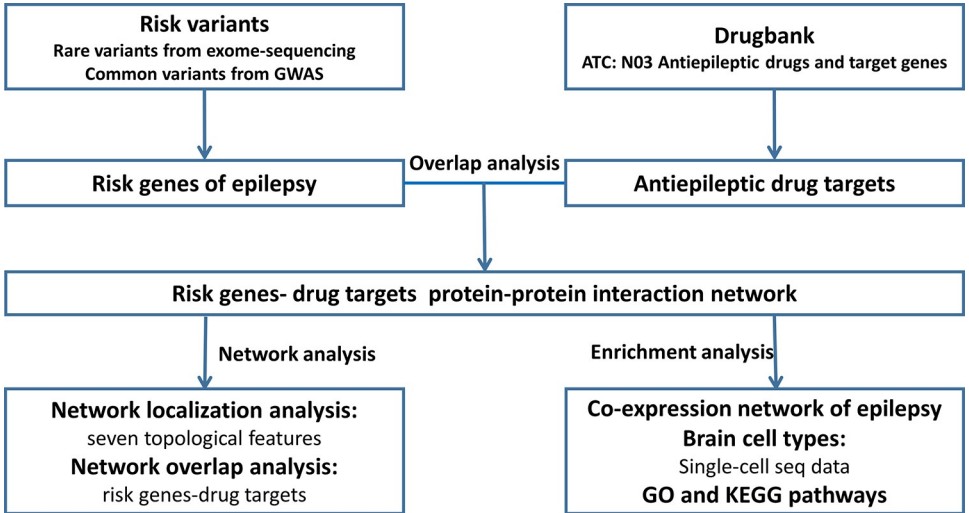

**Fig 1. The flowchart of our study.** GWAS: Genome-wide association study, GO: Gene Ontology, KEGG: Kyoto Encyclopedia of Genes and Genomes.

(version 4.1.0) to assess whether the targets of anti-epileptic drug were significantly over-represented in epilepsy risk genes.

## 3. Network topological features of PPI network generated by risk genes of epilepsy and anti-epilepsy drug targets

In order to investigate the network interactions between risk genes of epilepsy and anti-epilepsy drug targets, We first generated a PPI network using risk genes of epilepsy and anti-epileptic drug targets as input from a comprehensive PPI network data containing 17,252 genes and 471,448 experimental-validated interactions, which combined PPI networks from five databases, including Corum [17], BioPlex [18], CCSB [19], Integral [20] and BioGRID [21], then we calculated seven topological parameters (mean degree, all edges, largest subnetwork, closeness centrality, mean shortest distance, clustering coefficient and betweenness centrality) to evaluate the topological characteristics of this network. To evaluate the significance of these topological characteristics, we carried out permutation test by 5,000 times random sampling with the same number nodes as that in the observed network. This procedure was implemented by the network analysis software Network Calculator (network localization analysis module (https://github.com/Haoxiang-Qi/Network-Calculator.git) [22]. Network was visualized by Cytoscape Version 3.8.2 [23].

## 4. Network overlap analysis between risk genes of epilepsy and anti-epileptic drug targets

In order to evaluate whether there was significant interactions between anti-epileptic drug targets and epileptic risk genes at the network level, we calculated that mean shortest distance within network module of risk gene set of epilepsy (gene set A) as $d\_A$, mean shortest distance within network module of anti-epileptic drugs target genes set (gene set B) as $d\_B$, mean shortest distance between gene set A and gene set B as $d\_AB$, then we calculated the network proximity between A and B as $S\_AB = d\_AB - (d\_A + d\_B)/2$. Using the network overlap analysis module of the Network calculator [22], we evaluated the significance of network overlap by

permutation test of S_AB with a random sampling of 1,000 time using the same number of genes as that in gene set A and B.

## 5. Enrichment analysis of genes from PPI network in co-expression network of epilepsy

To evaluate whether genes identified by PPI network was enriched in co-expression network of epilepsy, we performed enrichment analysis of genes from our PPI network on a co-expression network of epilepsy including 320 genes, which was identified by gene co-expression network analysis (Weighted Gene Co-expression Network Analysis (WGCNA) [24] and DiffCoEx [25] in the brain reported by a previous study [26]. Fisher's Exact Test were performed with R (version 4.1.0).

## 6. Expression Weighted Cell Type Enrichment analysis of interaction network analysis between anti-epileptic drug targets and risk genes of epilepsy

To explore whether interaction network between target genes of anti-epileptic and drug risk genes of epilepsy could map on specific brain cell types, we implemented Expression Weighted Cell type Enrichment (EWCE) method, which used single-cell transcriptome dataset to calculate whether the average expression levels of input gene list was significantly stronger than that in randomly generated gene list with the same size as input in each annotated cell type [27]. Moreover, we utilized a superset of brain scRNA-seq data from the Karolinska Institutet (KI, S1 File) [11, 28–30], which included a total of 9,970 cells annotated with 24 cell types from mouse brain regions of neocortex, hippocampus, hypothalamus, striatum and midbrain, as well as samples enriched for oligodendrocytes, dopaminergic neurons and cortical parvalbuminergic interneurons. Since we used the mouse single-cell transcriptome sequencing data set as the background gene set, we first converted the human interaction network genes into mouse gene form, then we perform EWCE to calculate that significance of expression enrichment for interaction network genes in each brain cell type with 100,000 permutations and Bonferroni adjusted-P-value < 0.05 was considered as significance. R Package EWCE was utilized to perform the analysis and ggplot2 was used to generate graphs [31].

## 7. Gene ontology and kyoto encyclopedia of genes and genomes enrichment analysis of interaction network between anti-epileptic drug targets and risk genes of epilepsy

We used R Package clusterProfiler [32] to perform functional enrichment analysis for interaction network between anti-epileptic drug targets and risk genes of epilepsy, in which Gene ontology (GO) [33] functional annotation and kyoto encyclopedia of genes and genomes (KEGG) [34] annotation were used and Hypergeometric test was performed, with false discovery rate (FDR) < 0.01 as significance. R package ggplot2 was used to generate graphs [35].

## Results

### 1. Identification of risk genes of Epilepsy

Through literature searches, risk genes of epilepsy were obtained, and the results are indicated in S1 Table. A total of 118 epileptic risk genes including 102 rare variation genes and 16 newly discovered common variation genes were summed up.

## 2. Identification of targets of anti-epileptic drugs

After searching in DrugBank [16], we retrieved a total of 47 anti-epileptic drugs and 151 targets (S2 Table), of which identified 17 target genes (*CACNA1A*, *CHRNA4*, *CHRNA7*, *GABRA1*, *GABRA2*, *GABRB2*, *GABRG2*, *GRIK1*, *GRIN1*, *GRIN2B*, *KCNQ2*, *KCNQ3*, *SCN1A*, *SCN2A*, *SCN3A*, *SCN8A* and *SCN9A*) were overlapped with risk genes of epilepsy. Fisher's Exact Test demonstrated a significant over-representation of targets of anti-epileptic drug in the risk genes of epilepsy (odds ratio [OR] = 25.08, P = 2.2 ×10$^{-16}$).

## 3. Analysis of PPI network and network characteristics between epilepsy risk genes and anti-epileptic drug target genes

By network construction, we obtained a PPI network containing 247 interactions, including 55 epileptic risk genes and 94 anti-epileptic drug target genes (Fig 2). As shown in Table 1 and Fig 3, by analyzing seven topological characteristics of the PPI network, we identified that six of them were significant, in which all edges, mean degree, largest subnetwork, closeness centrality and clustering coefficient were significantly larger than randomly generated, and mean shortest distance was significantly smaller than randomly generated.

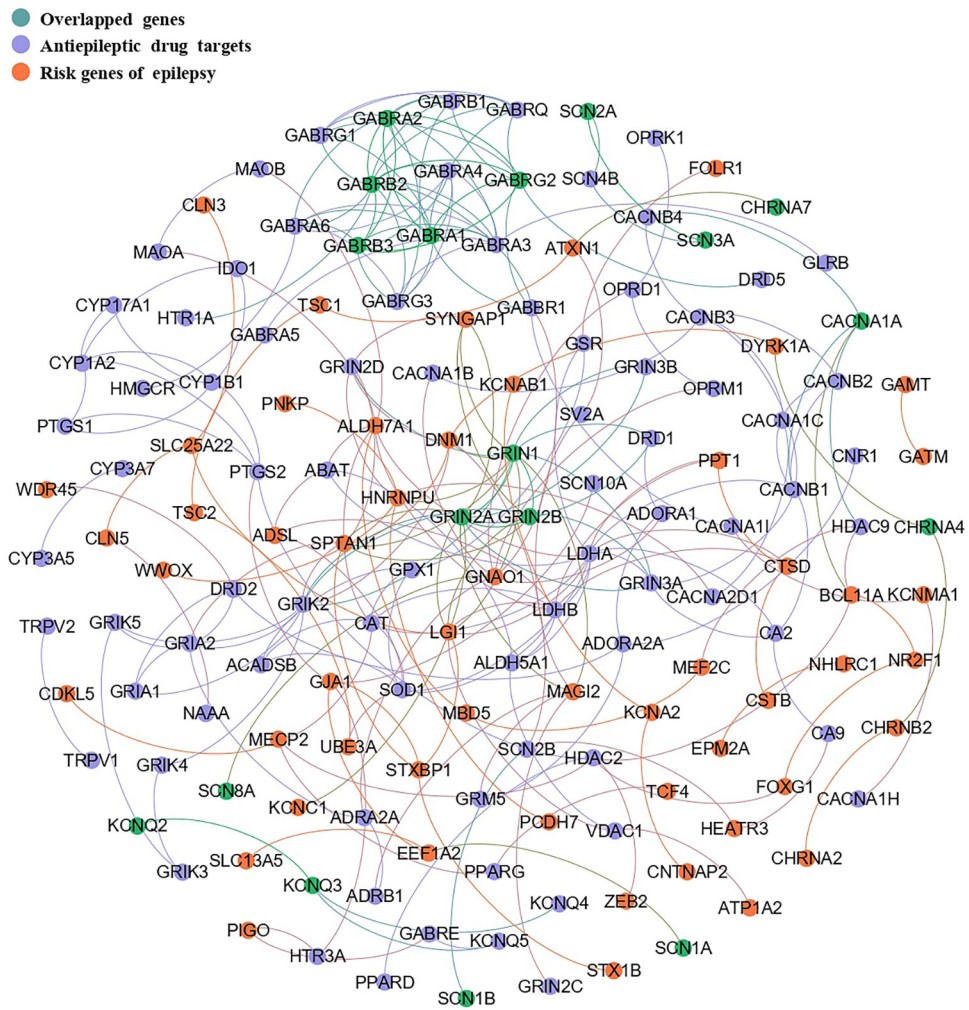

**Fig 2. The PPI network of risk genes of epilepsy and anti-epileptic drug target genes.**

**Table 1. Analysis of network characteristics between epilepsy risk genes and anti-epileptic drug target genes.**

| Network parameters | Observed | Random_mean | P_value |
| --- | --- | --- | --- |
| ALL edges | 247 | 87.648 | 0.000 |
| Largest subnetwork | 133 | 52.294 | 0.040 |
| Mean degree | 1.05 | 0.373 | 0.000 |
| Closeness centrality | 0.058 | 0.017 | 0.018 |
| Clustering coefficient | 0.134 | 0.028 | 0.011 |
| Betweenness centrality | 0.007 | 0.028 | 0.930 |
| Mean shortest distance | 1.38 | 1.655 | 0.011 |

## 4. Network overlap analysis between risk genes of epilepsy and anti-epileptic drug targets

To evaluate whether the network interactions between anti-epileptic drug targets and epileptic risk genes were significant, we performed network overlap analysis and the results showed mean shortest distance of network interactions among risk genes of epilepsy d_A was 1.47, that among anti-epileptic drug target genes d_B was 1.44, that between risk genes of epilepsy and anti-epileptic drug target genes d_AB was 1.37, and the network proximity between A and B (S_AB) was -0.09, which was significant with P-value of 0.04 calculated by permutation test (Fig 3H), demonstrating a significant interactome overlap between risk genes of epilepsy and anti-epileptic drug target genes.

## 5. Enrichment analysis in co-expression network of epilepsy

We performed enrichment analysis of 147 genes from our PPI network on 320 genes from a co-expression network of epilepsy [26] (S3 Table), and there were 22 genes (*CACNA1C*, *CACNB2*, *CACNB4*, *CHRNB2*, *DNM1*, *EEF1A2*, *GABRA1*, *GABRA3*, *GABRA4*, *GABRB2*, *GABRB3*, *GABRG2*, *GRIN1*, *KCNA2*, *KCNC1*, *KCNQ3*, *SCN1A*, *SCN4B*, *SCN8A*, *SOD1*, *STXBP1*, and *SV2A*) from PPI network were enriched in the co-expression network of epilepsy, and Fisher's Exact Test demonstrated the enrichment was significant (odds ratio [OR] = 11.80, P = $1.30 \times 10^{-15}$).

## 6. Brain cell-type enrichment analysis of interaction network between anti-epileptic drug targets and risk genes of epilepsy

To evaluated whether genes in interacted network of anti-epileptic drug targets and risk genes of epilepsy was significantly enriched in specific brain cell types, we performed EWCE in mouse brain scRNA-seq of Karolinska Institute (KI) dataset [11, 28–30]. For KI dataset, among 24 cell types, interacted network between anti-epileptic drug targets and risk genes of epilepsy were significantly enriched in four brain cell types (interneurons, Medium Spiny Neuron, CA1 pyramidal Neuron, and Somatosensory pyramidal Neuron), with Bonferroni-adjusted P-value < 0.05 (Fig 4 and S4 Table).

## 7. Gene ontology and kyoto encyclopedia of genes and genomes enrichment analysis of interaction network between anti-epileptic drug targets and risk genes of epilepsy

By Gene ontology (GO) and kyoto encyclopedia of genes and genomes (KEGG) pathway enrichment analysis of interaction network between anti-epileptic drug targets and risk genes of epilepsy, a total of 30 and 28 pathways were significantly enriched respectively with

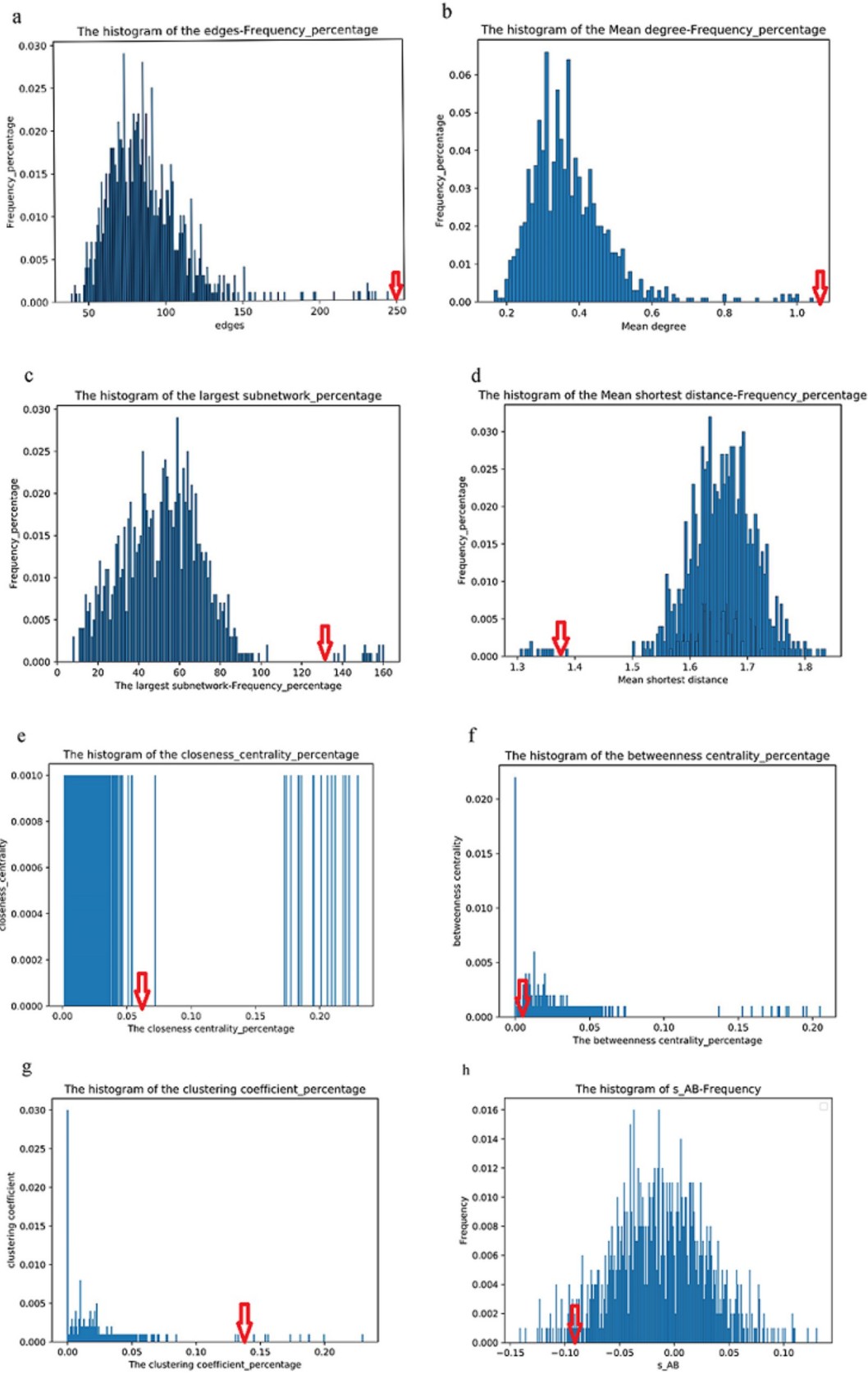

**Fig 3. Distribution of random sampling of network topological features.**

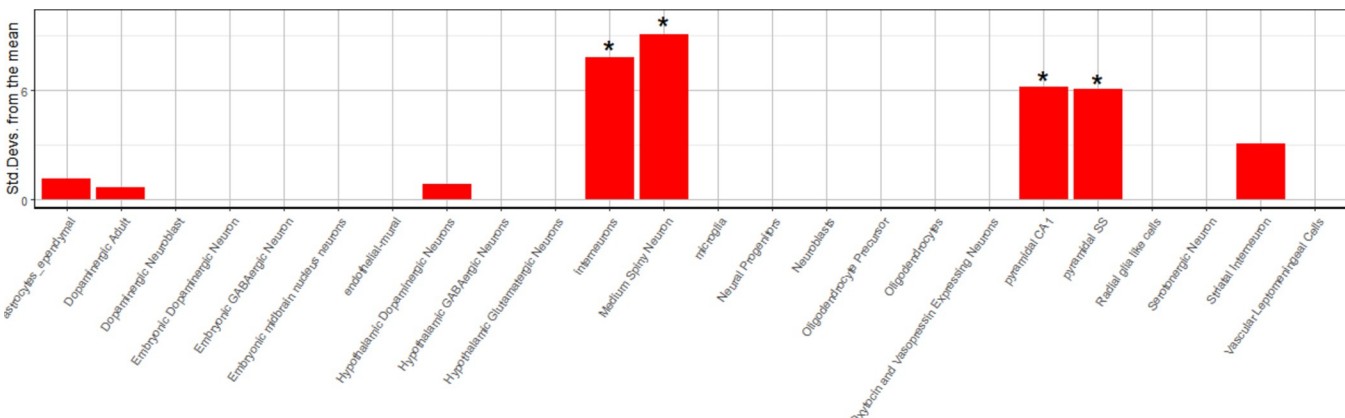

**Fig 4. Brain cell-type enrichment analysis of interaction network between anti-epileptic drug targets and risk genes of epilepsy.** *Bonferroni adjusted-P-value <0.05.

FDR < 0.01(Fig 5). Among these pathways, regulation of postsynaptic membrane potential, regulation of ion trans-membrane transduction, membrane depolarization, postsynaptic chemical synaptic transmission, and GABAergic Synaptic transmission are top significant for GO terms, as well as nicotine addiction, neuroactive ligand-receptor interaction, GABAergic synapse, morphine addiction and retrograde endocannabinoid signaling for KEGG.

## Discussion

Epilepsy is a widespread chronic nervous system disease, which affects about 70 million people all over the world [36], during this decade, with the development of sequencing technology, hundreds of risk genes associated with epilepsy have been identified by genetic studies [2], GWAS [12], as well as sequencing [10]. Meanwhile, currently there were more than 40 anti-epileptic drugs based on various target genes used for clinical application [37]. To systematically explore the interactions between risk genes of epilepsy and anti-epileptic drug targets, we used a network-based approach to construct PPI network with interacted risk genes of epilepsy and anti-epileptic drug targets and evaluated the interactome overlap between them. By analyzing seven topological parameters of the PPI network, we identified interactions in the network were significantly higher than randomly generated network with the same size of nodes, similar results were observed in network of schizophrenia and related antipsychotic drugs [38], indicating network constructed by risk genes of epilepsy and anti-epileptic drug targets formed a distinct inner-connected network rather than randomly scattered in the interactome.

To investigate whether there was significant overlap between epilepsy risk genes with anti-epileptic drug targets, we identified 17 overlapped genes *(CACNA1A, CHRNA4, CHRNA7, GABRA1, GABRA2, GABRB2, GABRG2, GRIK1, GRIN1, GRIN2B, KCNQ2, KCNQ3, SCN1A, SCN2A, SCN3A, SCN8A and SCN9A)*, which showed a significant overlap by Fisher's Exact Test, demonstrating anti-epileptic drug targets were over-represented in risk genes of epilepsy. Meanwhile, we also identified there was significant interactome overlap between them (P = 0.04), suggesting the distance between genes of anti-epileptic drug targets and risk genes of epilepsy was significantly closer than the distance among genes in their respective networks. Similar results have been reported in previous studies investigating overlap between the drug targets of antipsychotics and schizophrenia risk genes [38, 39]. These results indicated the genetic overlap between the pathogenesis of epilepsy and the action mechanism of anti-

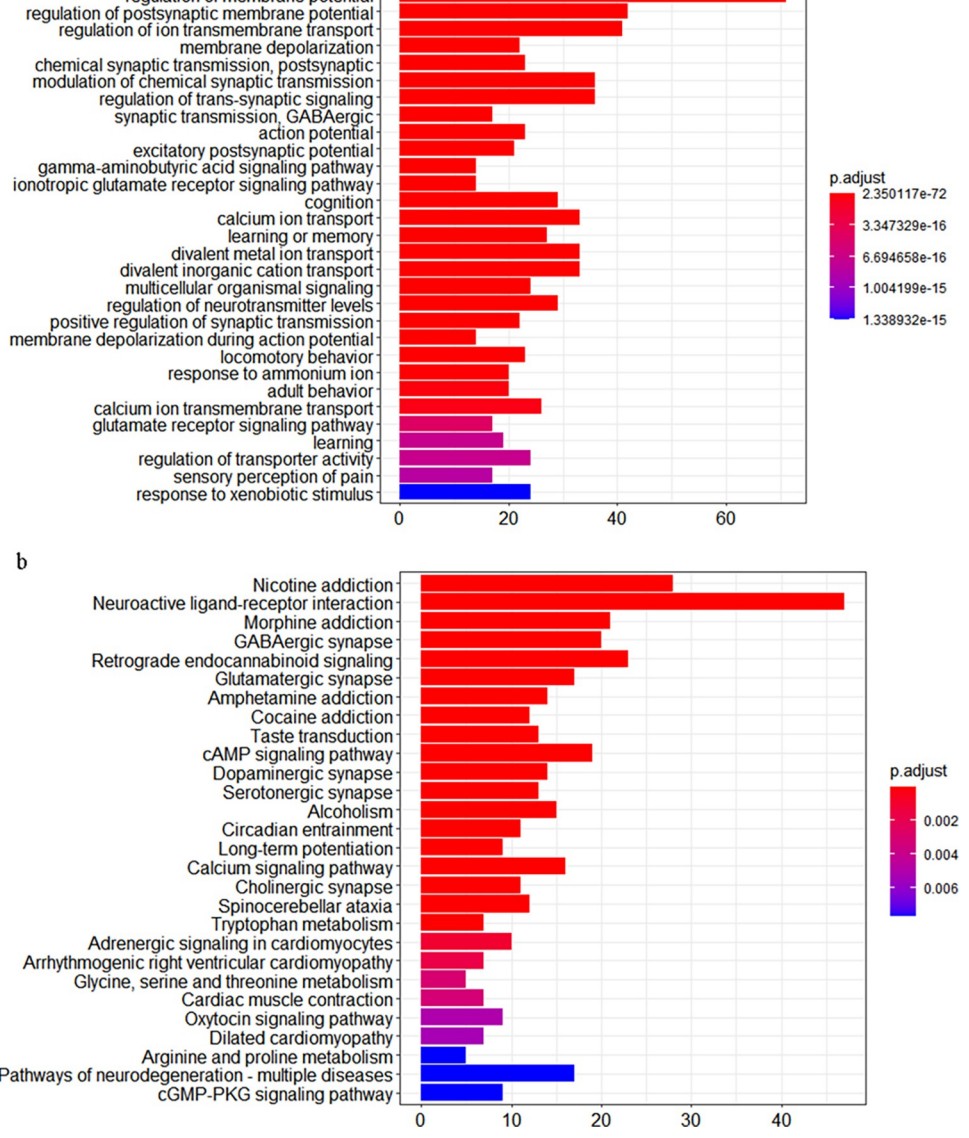

**Fig 5. The GO and KEGG pathway enrichment analysis of interaction network between anti-epileptic drug targets and risk genes of epilepsy.** *FDR<0.01.

epileptic drugs and provide genetic support evidence for the treatment of epilepsy with anti-epileptic drugs.

Moreover, when we compared genes in this PPI network with genes from a co-expression network of epilepsy [26] (S3 Table), we also identified a significant enrichment of genes in our PPI network in the co-expression network of epilepsy, with 22 gene overlapped, interestingly, all the 22 genes were anti-epileptic drug targets, in which 7 genes were also risk genes (*GABRA1*, *GABRB2*, *GABRG2*, *GRIN1*, *KCNQ3*, *SCN1A*, and *SCN8A*). Voltage-gated sodium channels (VGSCs) play a critical role in generation of action potentials, *SCN1A*, *SCN2A*, *SCN3A*, *SCN8A* and *SCN1B* have been identified to be associated with a spectrum of epilepsy phenotypes and neurodevelopmental disorders [40]. Besides, So far, the most widespread viewpoint considered that GABA$_A$ receptor, as an isomer receptor that binds to GABA, affects

the excitability of nerve cells by stimulating chloride ions influx into the postsynaptic membrane and exerts anti-epileptic effect [41]. It was also reported a personalized therapy in a *GRIN1* mutated girl with intellectual disability and epilepsy [42]. Our results demonstrated that it was important to know the functional effect (Loss-of-function versus Gain-of-function) of a variant for genes which were both risk genes and drug targets to orient therapeutic decisions.

By EWCE in brain scRNA-seq of Karolinska Institute (KI) dataset, including 24 cell types, we found network formed by anti-epileptic drug targets and risk genes of epilepsy were significantly enriched in four brain cell types (interneurons, Medium Spiny Neuron, CA1 pyramidal Neuron, and Somatosensory pyramidal Neuron), These results suggest that the pathogenesis of epilepsy might the result of impaired function of some specific brain cell types [43, 44] and anti-epileptic drugs may play a role in some specific brain cell types [45].

In our study, we explored the network interaction between anti-epileptic drug and risk genes of epilepsy by systematic data collection and integrative analysis. However, there were still some inevitable limitations. First, although we used interactome data by integrating five comprehensive PPI databases, there might still exist interactions between risk genes and drug targets that could not be identified by current interactome data, which are worthy of being explored with the update of interactome data. Second, since the number of cells taken in the single-cell data sets accounts for only a small portion of the whole brain tissue, they may not represent all types of brain cells and need further validation. Third, all the results were obtained by systems biology and network analyses based on data from public databases, which might only reveal underlying mechanisms with currently existing information and need further experimental validation.

## Conclusion

In this study, we systematically explored the interaction of epilepsy risk genes and anti-epileptic drug targets through a network-based approach. We identified a significantly localized PPI network with 55 epileptic risk genes and 94 anti-epileptic drug target genes, and network overlap analysis showed significant interactome overlap between risk genes and drug targets. Besides, cell type enrichment analysis indicated genes in this network were significantly enriched in 4 brain cell types (Interneuron, Medium Spiny Neuron, CA1 pyramidal Neuron, and Somatosensory pyramidal Neuron). These results provide evidence for interactions between epilepsy risk genes and anti-epileptic drug targets from the perspective of network biology.

## Supporting information

**S1 Table. The risks genes of epilepsy.**
(DOCX)

**S2 Table. The target genes of anti-epileptic drugs.**
(DOCX)

**S3 Table. Genes of co-expression network of epilepsy.**
(DOCX)

**S4 Table. Brain cell-types enrichment analysis of interaction network of risk gene of epilepsy and anti-epileptic drug targets.**
(DOCX)

**S1 File. Raw KI dataset for analysis of cell types enriched by anti-epileptic drugs.**
(ZIP)

## Author Contributions

**Data curation:** Yu-Qin Lv, Xing Wang.

**Resources:** Lei Gao, Jing-Jun Zhang.

**Supervision:** Lei Gao, Jing-Jun Zhang.

**Writing – original draft:** Yu-Qin Lv, Xing Wang, Yu-Zhuang Jiao, Yan-Hua Wang, Na Wang.

**Writing – review & editing:** Yu-Qin Lv, Xing Wang.

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
