## [Decision Letter · Decision Letter 0]

22 Apr 2022

PONE-D-22-05934Interactome overlap between risk genes of epilepsy and targets of antiepileptic drugsPLOS ONE

Dear Dr. Gao,

Thank you for submitting your manuscript to PLOS ONE. Given the positive response from Reviewer 1 and to avoid further delay, we have decided to move forward and invite you to revise your manuscript. Please see the reviewer's comments appended in this letter. We look forward to receiving your revised manuscript.

Sincerely,

Nien-Pei Tsai

We look forward to receiving your revised manuscript.

Kind regards,

Nien-Pei Tsai, PhD

Academic Editor

PLOS ONE

Journal Requirements:

"This research was supported by Medical Health Science and Technology Project of Shandong Provincial Health Commission (2019WS391), Academic Promotion Program of Shandong First Medical University (2019QL013), National Natural Science Foundation of China (32000477), Shandong provincial Natural Science Foundation (ZR2020MC061), Science and Technology Program of Colleges and Universities in Shandong Province(J15LL07) and the planned project of Tai'an Science and Technology Bureau(2017NS0248).

We note that you have provided funding information. However, funding information should not appear in the Acknowledgments section or other areas of your manuscript. We will only publish funding information present in the Funding Statement section of the online submission form. 

"This research was supported by Medical Health Science and Technology Project of Shandong Provincial Health Commission (2019WS391), Academic Promotion Program of Shandong First Medical University (2019QL013), National Natural Science Foundation of China (32000477), Shandong provincial Natural Science Foundation (ZR2020MC061), Science and Technology Program of Colleges and Universities in Shandong Province(J15LL07) and the planned project of Tai'an Science and Technology Bureau(2017NS0248)."

Reviewers' comments:

Reviewer's Responses to Questions

**Comments to the Author**

1. Is the manuscript technically sound, and do the data support the conclusions?

Reviewer #1: Yes

2. Has the statistical analysis been performed appropriately and rigorously? 

Reviewer #1: N/A

3. Have the authors made all data underlying the findings in their manuscript fully available?

Reviewer #1: No

4. Is the manuscript presented in an intelligible fashion and written in standard English?

Reviewer #1: No

5. Review Comments to the Author

Reviewer #1: The authors present an interesting study systematically investigating the

potential interaction of epilepsy risk genes and antiepileptic drug targets. Although the area and novelty of the study approach have the potential for publication, I have some major and minor concerns which are listed below:

Major Concerns:

1) Please briefly explain the process of study with a graphical abstract or a flowchart showing the pipe lines of your marker selections.

2) Did you used any ethical analysis due to genetic risk factors? How you exclude the normal variations in each population? Any Haplogroup analysis were conducted?

3) What is the rationale behind choosing each data bae? Please briefly explain it for each data base.

4) It seems that false discovery rate (FDR) < 0.01 used for multiple correction. While the Bonferroni multiple correction test is more conservative why you didn’t use Bonferroni? Is that possible that some of those markers excluded if you use more conservative tests?

5) I was wondering that why you didn’t used the WGCNA R software package to analysis of weighted gene co-expression network analysis (WGCNA)? Please explain the reason in manuscript or use the analysis and add the results in your revised manuscript.

6) Discussion part should explain the related genetic markers much more mechanistically to help the readers understand why future treatments may us these data to choose more targeted drug compounds.

7) Please add a paragraph explaining the limitations of the study.

Minor concerns:

1(Use abbreviations correctly in whole manuscript.

2) There are other minor issues (such as diction, editing, and linguistic errors) that need to be corrected in the revised version of the MS. It is highly recommended that a native English speaker edits the entire manuscript (particularly the introduction section) before submission of the revised MS.

3) In line 113, please indicate the version of Cytoscape software that you used.

6. PLOS authors have the option to publish the peer review history of their article (what does this mean?). If published, this will include your full peer review and any attached files.

Reviewer #1: **Yes: **Arvin Haghighatfard

---

## [Author Response · Author response to Decision Letter 0]

14 Jun 2022

Response Letter

Dear editor and reviewers:

Thank you for your reviewing! We have revised our manuscript according to your suggestion, and our response was as follows:

Comments to the Author

1. Is the manuscript technically sound, and do the data support the conclusions?

Reviewer #1: Yes

2. Has the statistical analysis been performed appropriately and rigorously?

Reviewer #1: N/A

3. Have the authors made all data underlying the findings in their manuscript fully available?

Reviewer #1: No

Response: We have added all data underlying the findings in supporting information (Supplementary Table 1-4) and data availability statement.

4. Is the manuscript presented in an intelligible fashion and written in standard English?

Reviewer #1: No

Response: We have checked our manuscript and revised them according to your suggestion. 

5. Review Comments to the Author

Reviewer #1: The authors present an interesting study systematically investigating the

potential interaction of epilepsy risk genes and antiepileptic drug targets. Although the area and novelty of the study approach have the potential for publication, I have some major and minor concerns which are listed below:

Major Concerns:

1)Please briefly explain the process of study with a graphical abstract or a flowchart showing the pipe lines of your marker selections.

Response: We have added a flowchart of the pipelines according to your suggestions as Figure 1.

2)Did you used any ethical analysis due to genetic risk factors? How you exclude the normal variations in each population? Any Haplogroup analysis were conducted?

Response: We obtained risk genes of epilepsy identified by both common variants and rare variants, of which common variants are extracted from the largest trans-ethnic meta-analyses of genome-wide association studies currently[12], which included 15,212 individuals with epilepsy and 29,677 controls and identified 16 genome-wide significant loci with P-value<5.0×10-8. Rare variants were identified by exome-sequencing under the largest sample size of 1165 cases and 3877 controls [13], as the article demonstrated, these variants in normal population have been investigated in the article and demonstrated that they are exceptionally rare in the general population. 

3)What is the rationale behind choosing each data bae? Please briefly explain it for each data base.

Response:

1.DisGeNET database[14] and MalaCards database[15] are comprehensive databases with disease associated genes and variants, which provided evidence of risk genes.

2.DrugBank[16] is a web-based database containing comprehensive molecular information about drugs, their mechanisms of action, interactions, and their targets, it provided drug targets information with literature supported evidence.

3.Corum[17], BioPlex[18], CCSB[19], Integral[20] and BioGRID[21] are protein-protein interaction databases that are commonly used in network analysis, to comprehensively capture the PPI data, we integrated five PPI database together. 

4.KI dataset [28-30] is a superset of brain scRNA-seq data from the Karolinska Institutet (KI) including 24 cell types from brain regions of neocortex, hippocampus, hypothalamus, striatum and midbrain, which has been used to identify enrichment of risk genes in major brain disorders [11,27].

4)It seems that false discovery rate (FDR) < 0.01 used for multiple correction. While the Bonferroni multiple correction test is more conservative why you didn’t use Bonferroni? Is that possible that some of those markers excluded if you use more conservative tests?

Response: For Expression Weighted Cell type Enrichment (EWCE), we have revised the multiple correction method to Bonferroni multiple correction test, under this threshold, genes were significantly enriched in four brain cell types (interneurons, Medium Spiny Neuron, CA1 pyramidal Neuron, Somatosensory pyramidal Neuron), and we revised the results in “6. Brain cell-type enrichment analysis of interaction network between antiepileptic drug targets and risk genes of epilepsy”, Fig 3 and Supplementary Table 4. For pathway enrichment analysis, FDR was the most commonly used method, if Bonferroni multiple correction test was used, meaningful pathways might be excluded.

5)I was wondering that why you didn’t used the WGCNA R software package to analysis of weighted gene co-expression network analysis (WGCNA)? Please explain the reason in manuscript or use the analysis and add the results in your revised manuscript.

Response: Our study was based on interactome data, which was not expression data. However, according to your valuable suggestion, we used a co-expression network of 320 genes (M30) related with epilepsy, which was obtained by gene co-expression network analysis (WGCNA [24] and DiffCoEx[25]) in the brain reported by a previous study[26]. We identified 22 target genes (CACNA1A,CHRNA4,CHRNA7,GABRA1,GABRA2,GABRB2,GABRG2,GRIK1,GRIN1,GRIN2B,KCNQ2,KCNQ3,SCN1A,SCN2A, SCN3A,SCN8A and SCN9A) were overlapped with risk genes of epilepsy.. Fisher’s Exact Test demonstrated a significant over-representation of targets of antiepileptic drug in the risk genes of epilepsy (odds ratio [OR] =11.80, P=1.30 ×10 −15). We have revised them in related Methods and Results section.

6)Discussion part should explain the related genetic markers much more mechanistically to help the readers understand why future treatments may us these data to choose more targeted drug compounds.

Response: we have add a paragraph in “Discussion” in the manuscript as:

“Moreover, when we compared genes in this PPI network with genes from a co-expression network of epilepsy [26] (Supplementary Table 3), we also identified a significant enrichment of genes in our PPI network in the co-expression network of epilepsy, with 22 gene overlapped, interestingly, all the 22 genes were antiepileptic drug targets, in which 7 genes were also risk genes (GABRA1, GABRB2, GABRG2, GRIN1, KCNQ3, SCN1A, and SCN8A). Voltage-gated sodium channels (VGSCs) play a critical role in generation of action potentials, SCN1A, SCN2A, SCN3A, SCN8A and SCN1B have been identified to be associated with a spectrum of epilepsy phenotypes and neurodevelopmental disorders[40]. Besides, So far, the most widespread viewpoint considered that GABAA receptor, as an isomer receptor that binds to GABA, affects the excitability of nerve cells by stimulating chloride ions influx into the postsynaptic membrane and exerts antiepileptic effect[41]. It was also reported a personalized therapy in a GRIN1 mutated girl with intellectual disability and epilepsy [42]. Our results demonstrated that it was important to know the functional effect (Loss-of-function versus Gain-of-function) of a variant for genes which were both risk genes and drug targets to orient therapeutic decisions.”

7)Please add a paragraph explaining the limitations of the study.

Response: we have add a paragraph before “Conclusion” in the manuscript as:

“In our study, we explored the network interaction between antiepileptic drug and risk genes of epilepsy by systematic data collection and integrative analysis. However, there were still some inevitable limitations. First, although we used interactome data by integrating five comprehensive PPI databases, there might still exist interactions between risk genes and drug targets that could not be identified by current interactome data, which are worthy of being explored with the update of interctome data. Second, since the number of cells taken in the single-cell data sets accounts for only a small portion of the whole brain tissue, they may not represent all types of brain cells and need further validation. Third, all the results were obtained by systems biology and network analyses based on data from public databases, which might only reveal underlying mechanisms with currently existing information and need further experimental validation.” 

Minor concerns:

1(Use abbreviations correctly in whole manuscript.

Response: We have checked our manuscript and revised them according to your suggestion.

2)There are other minor issues (such as diction, editing, and linguistic errors) that need to be corrected in the revised version of the MS. It is highly recommended that a native English speaker edits the entire manuscript (particularly the introduction section) before submission of the revised MS.

Response:We have checked our manuscript and revised them according to your suggestion.

3)In line 113, please indicate the version of Cytoscape software that you used.

Response: Cytoscape Version 3.8.2 was used.

---

## [Decision Letter · Decision Letter 1]

20 Jul 2022

Interactome overlap between risk genes of epilepsy and targets of antiepileptic drugs

PONE-D-22-05934R1

Dear Dr. Gao,

We’re pleased to inform you that your manuscript has been judged scientifically suitable for publication and will be formally accepted for publication once it meets all outstanding technical requirements.

Kind regards,

Nien-Pei Tsai, PhD

Academic Editor

PLOS ONE

Additional Editor Comments (optional):

Reviewers' comments:

Reviewer's Responses to Questions

**Comments to the Author**

1. If the authors have adequately addressed your comments raised in a previous round of review and you feel that this manuscript is now acceptable for publication, you may indicate that here to bypass the “Comments to the Author” section, enter your conflict of interest statement in the “Confidential to Editor” section, and submit your "Accept" recommendation.

Reviewer #1: All comments have been addressed

2. Is the manuscript technically sound, and do the data support the conclusions?

Reviewer #1: (No Response)

3. Has the statistical analysis been performed appropriately and rigorously? 

Reviewer #1: Yes

4. Have the authors made all data underlying the findings in their manuscript fully available?

Reviewer #1: Yes

5. Is the manuscript presented in an intelligible fashion and written in standard English?

Reviewer #1: Yes

6. Review Comments to the Author

Reviewer #1: The comments had been addressed in the revised manuscript. But I still strongly suggest an English language editing in the whole manuscript before the publication.

7. PLOS authors have the option to publish the peer review history of their article (what does this mean?). If published, this will include your full peer review and any attached files.

Reviewer #1: **Yes: **Arvin Haghighatfard

---

## [Editor Report · Acceptance letter]

16 Aug 2022

PONE-D-22-05934R1 

Interactome overlap between risk genes of epilepsy and targets of anti-epileptic drugs 

Dear Dr. Gao:

I'm pleased to inform you that your manuscript has been deemed suitable for publication in PLOS ONE. Congratulations! Your manuscript is now with our production department. 

Kind regards, 

on behalf of

Dr. Nien-Pei Tsai 

Academic Editor

PLOS ONE